# Development and Validation of a Good Manufacturing Process for IL-4-Driven Expansion of Chimeric Cytokine Receptor-Expressing CAR T-Cells

**DOI:** 10.3390/cells10071797

**Published:** 2021-07-15

**Authors:** May C. I. van Schalkwyk, Sjoukje J. C. van der Stegen, Leticia Bosshard-Carter, Helen Graves, Sophie Papa, Ana C. Parente-Pereira, Farzin Farzaneh, Christopher D. Fisher, Andrew Hope, Antonella Adami, John Maher

**Affiliations:** 1Guy’s Cancer Centre, School of Cancer and Pharmaceutical Sciences, King’s College London, Great Maze Pond, London SE1 9RT, UK; May.VanSchalkwyk@lshtm.ac.uk (M.C.I.v.S.); StegenS@mskcc.org (S.J.C.v.d.S.); l.bosshard-carter@nhs.net (L.B.-C.); sophie.papa@kcl.ac.uk (S.P.); anacatpp@gmail.com (A.C.P.-P.); antonella.adami@kcl.ac.uk (A.A.); 2Immune Monitoring Laboratory, Clinical Research Facility, NIHR Biomedical Research Centre at Guy’s and St Thomas’ NHS Foundation Trust and King’s College London, Great Maze Pond, London SE1 9RT, UK; helen@alchemab.com; 3Guy’s and St Thomas’ NHS Foundation Trust, Department of Medical Oncology, Great Maze Pond, London SE1 9RT, UK; 4The Rayne Institute, School of Cancer and Pharmaceutical Sciences, King’s College London, London SE5 9NU, UK; farzin.farzaneh@kcl.ac.uk; 5Good Manufacturing Practice Unit, Clinical Research Facility, NIHR Biomedical Research Centre at Guy’s and St Thomas’ NHS Foundation Trust and King’s College London, Great Maze Pond, London SE1 9RT, UK; c.fisher@autolus.com (C.D.F.); drewhope@hotmail.co.uk (A.H.); 6Department of Immunology, Eastbourne Hospital, Kings Drive, Eastbourne BN21 2UD, UK; 7Department of Clinical Immunology and Allergy, King’s College Hospital NHS Foundation Trust, Denmark Hill, London SE5 9RS, UK; 8Leucid Bio Ltd., Guy’s Hospital, Great Maze Pond, London SE1 9RT, UK

**Keywords:** good manufacturing practice, closed process, chimeric antigen receptor, retrovirus, interleukin-4

## Abstract

Adoptive cancer immunotherapy using chimeric antigen receptor (CAR) engineered T-cells holds great promise, although several obstacles hinder the efficient generation of cell products under good manufacturing practice (GMP). Patients are often immune compromised, rendering it challenging to produce sufficient numbers of gene-modified cells. Manufacturing protocols are labour intensive and frequently involve one or more open processing steps, leading to increased risk of contamination. We set out to develop a simplified process to generate autologous gamma retrovirus-transduced T-cells for clinical evaluation in patients with head and neck cancer. T-cells were engineered to co-express a panErbB-specific CAR (T1E28z) and a chimeric cytokine receptor (4αβ) that permits their selective expansion in response to interleukin (IL)-4. Using peripheral blood as starting material, sterile culture procedures were conducted in gas-permeable bags under static conditions. Pre-aliquoted medium and cytokines, bespoke connector devices and sterile welding/sealing were used to maximise the use of closed manufacturing steps. Reproducible IL-4-dependent expansion and enrichment of CAR-engineered T-cells under GMP was achieved, both from patients and healthy donors. We also describe the development and approach taken to validate a panel of monitoring and critical release assays, which provide objective data on cell product quality.

## 1. Introduction

Adoptive immunotherapy using chimeric antigen receptor (CAR) engrafted T-cells holds great promise as a novel and effective approach to the treatment of cancer. Six CAR T-cell products have now been approved in various countries worldwide for the treatment of B-cell and plasma cell malignancies. However, many obstacles need to be overcome to enable the widespread application of this emerging therapeutic modality. One key hurdle relates to the technical challenges associated with manufacture of cell products, leading to poor standardization and variability in quality [1,2]. Most commonly, the starting material is obtained using a leukapheresis procedure, which is widely considered to be an essential step in the manufacturing process [3]. However, leukapheresis scheduling can prove challenging since it is used for several additional applications, including the removal of malignant leukocytes, harvesting of stem cells for transplantation and treatment of selected non-malignant indications such as inflammatory bowel disease. Thereafter, CAR T-cells are expanded using complex and costly cell processing methodologies in highly specialized good manufacturing practice (GMP) facilities [4,5,6]. It is preferred that protocols employ closed processing in order to minimise risk of contamination, although open processing steps remain in widespread use [7]. Increasingly, there has been a move towards the use of automated systems with a view to upscaling manufacture to industrial levels [7].

Mindful of these challenges, our aim was to design a simple, maximally closed and robust sterile production protocol for gene-modified T-cells that could be implemented in a newly established academic GMP manufacturing facility in the United Kingdom. We undertook this work in preparation for a first in man clinical trial of “T4 immunotherapy” in patients with head and neck squamous cell carcinoma (HNSCC; ClinicalTrials.gov number, NCT01818323; EudraCT Number: 2012-001654-25, accessed on 15 July 2021). Owing to limited capacity for leukapheresis, we opted to use whole blood as starting material. Moreover, expanded CAR T-cells were formulated as a fresh rather than a cryopreserved drug product. In the preparation of T4 immunotherapy, T-cells are engineered to express a CAR named T1E28z that directs specificity against several pathogenetically relevant ErbB dimers in HNSCC [8]. Target recognition is achieved using the T1E peptide, which is a promiscuous chimeric ErbB ligand derived from transforming growth factor-α and epidermal growth factor [9]. Expansion of T1E28z^+^ T-cells is conveniently achieved by co-expression with 4αβ—a chimeric cytokine receptor in which the ectodomain of the IL-4 receptor α subunit has been joined to the transmembrane and endodomain of the shared IL-2/15 receptor b chain [10]. Addition of IL-4 to T-cells that co-express T1E28z and 4αβ (a combination designated “T4”) promotes the selective enrichment of gene-modified cells [10], which we anticipated would enhance the robustness of cell product manufacture.

Several approaches have been integrated here to design a process that simplifies manufacture and minimises open processing steps reliant on microbiologically-free clean air, thereby reducing isolator use. The approach is generic and may be adapted for use in related applications using either T-cell receptor or CAR engrafted T-cells. Procedures used to develop and validate this manufacturing process in compliance with GMP are presented. In addition, we describe the development of a set of monitoring and critical release assays which provide objective and recordable information regarding cell product quality.

## 2. Materials and Methods

### 2.1. Description of Manufacturing Process

Assignment of terminology used in the development of this manufacturing process is shown in Figure 1 and was based upon EU Commission Directive 2009/120/EC: http://ec.europa.eu/health/files/eudralex/vol-1/dir_2009_120/dir_2009_120_en.pdf (accessed on 21 May 2013). This terminology was used to structure the Investigational Medicinal Product Dossier (IMPD) for T4 immunotherapy.

A schematic overview of the manufacturing process is provided in Figure 2. Sterile materials and procedures were used throughout the process. Settle (settling) plates were in use throughout manufacturing while operating in a Grade D room. Similarly, settle plates were placed inside a grade A isolator when in use. While operating inside the isolator, the inbuilt alarmed particle counter monitor was turned on. Should the alarm activate, all activities were ceased until the particle count had returned to zero. Finger dabs were performed at the end of each procedure. Where indicated, sterile welding and sealing was performed using a TSCD^®^ Sterile Tubing Welder and T-SEAL^®^ II Tube Sealing Device (Terumo BCT, Lakewood, CO, USA). The wafers used by the device heat up to 300 °C, maintaining sterility throughout. All bags, tubing, serum and viral vector used in this procedure were both sterile and endotoxin-free, as specified by the manufacturers.

#### 2.1.1. Pre-Production of Materials in a Grade a Isolator (Day-1; Figure 2A)

On the day before blood was collected, medium aliquots, cytokine aliquots and a RetroNectin^®^-coated bag were prepared inside a grade A isolator (Amercare, Thame, UK) that had been sterilized with hydrogen peroxide. All raw materials were sprayed with Klercide™ 70/30 denatured ethanol blended with deionized water, wiped down with Klerwipe™ 70/30 isopropyl alcohol blended with deionized water and stored in a dedicated container prior its use. Upon transfer of raw material to a grade D facility room, raw material was sprayed and wiped a second time as above and then transferred into the grade A isolator via the transfer hatch:

(i) Medium: To prepare complete medium, 10% human AB serum (non-heat inactivated; Seralab, Haywards Heath, UK) and 10% stable glutamine ATMP ready (PAA, Yeovil, UK) were added to four 1L bottles of X-VIVO™ 15 medium (BE02-054Q; Lonza, Slough, UK).

Pooled human serum used in this trial was procured from healthy donors in paid collection facilities located in the United States in compliance with FDA Code of Federal Regulation Title 21. Batches were issued with a satisfactory certificate of analysis.

Media additions are made via Mini-Spikes^®^ (B. Braun, Sheffield, UK) which were inserted into the cap of bottles containing glutamine, serum and media, allowing the use of needle-free Luer syringes (BD Biosciences, Oxford, UK) to transfer solutions to the media bottle. Complete medium was then transferred into four sterile 1000 mL transfer bags (Terumo UK, Egham, UK) using a custom-made lid and tubing set (Figure 3A; BioPharma Dynamics, Didcot, UK).

(ii) Cytokines: Interleukin (IL)-2 (Clinical grade for human administration; Aldesleukin; Clinigen, London, UK) and IL-4 (GMP grade; Miltenyi Biotec, Bisley, UK) were dissolved in complete medium. Three mL aliquots containing desired cytokine quantities (IL-2 5000 IU × 3 aliquots; IL-4 5 μg × 4 aliquots; 10 μg × 4 aliquots; 20 μg × 4 aliquots; 40 μg × 2 aliquots) were transferred to sterile sample pouches (Cytiva, Eysins, Switzerland) using Luer syringes. A 10 mL Luer syringe was filled with sterile air within the grade A isolator and was then attached to the pouch (Figure 3B) to allow transfer of content when pouch is welded to the cell culture. Aliquoted media and cytokines were stored at 4 °C until use.

(iii) RetroNectin^®^-pre-coated bag: A PermaLife™ PL240 cell culture bag (Origen Biomedical, Austin, TX, USA) was pre-coated with GMP-grade RetroNectin^®^ (Takara Bio Europe, Saint-Germain-en-Laye, Paris, France) to facilitate gene transfer. RetroNectin^®^ (2.5 mg in 2.5 mL) was added to 150 mL of saline (0.9 % *w*/*v*) and then transferred by Luer syringe into the PermaLife™ bag. After gentle turning to facilitate mixing, the RetroNectin^®^-coated bag was stored at 4 °C until required on day 3.

#### 2.1.2. Isolation and Activation of Patient-Derived T-Cells (Days 1–2; Figure 2B)

##### Preparation of Sepax Cell Separation Kit

Working in a grade A isolator, a Sepax tubing set kit (Cytiva) was pre-loaded with ficoll (Ficoll Paque PLUS 100 mL vialFF, Cytiva) and transferred out for assembly onto a Sepax C-Pro instrument (Cytiva).

##### Isolation of Peripheral Blood Mononuclear Cells

Blood (130 mL target volume) was collected by venipuncture into a clinical blood pack containing citrate phosphate dextrose adenine (CDPA) anticoagulant. A patient identification label was attached which specifies: First and Last Name, Date of Birth, Hospital Number, Batch Number (referring to the CAR T-cell batch that will be produced), Date, Trial Subject ID and Blood Volume. Mononuclear cells are stable in anticoagulated blood for at least 6 h when stored at room temperature [11]. For this reason, harvested blood must be processed within this interval. A 5mL sample was taken from the filled transfusion bag in order to test for sterility using the BactT/ALERT^®^ microbial detection system (bioMérieux, Basingstoke, UK).

Peripheral blood mononuclear cells (PBMC) were isolated using the NeatCell protocol on a Sepax^®^ C-Pro (Cytiva) closed automated cell separation platform. The blood bag was welded to the input line of a single-use Sepax cell separation kit along with the washing resuspension solution bag consisting of complete medium containing CPDA anticoagulant. Using the “small final volume” option (NeatCell), PBMC are removed from the chamber in 10 mL of complete medium within a 20 mL syringe connected to the manifold. Using sterile welding, PBMC were transferred to a MACS GMP cell expansion bag (Miltenyi Biotec). Cell count was determined using a Scepter™ automated cell counter (Millipore, Billerica, MA, USA), enumerating events within the PBMC gate. Complete medium was added to achieve a final cell concentration of 3 × 10^6^ cells/mL using the following closed method, incorporating sterile welding and sealing steps. A Luer syringe with tube (Quest/Origen Biomedical) was welded to a 1 L transfer bag containing complete medium using a tube welder TSCD-II (Terumo BCT). The required amount of culture medium was withdrawn into the syringe and the tubing then closed using a T-SEAL II sterile sealing device (Terumo BCT), thereby disconnecting the syringe-extension from the transfer bag. The syringe with tube was next welded to the PBMC bag, allowing medium transfer. Finally, the syringe with tube was removed by sterile sealing.

##### Preparation and Addition of T-Cell Activation Beads

Working in a grade A isolator, CD3/CD28 CTS™ paramagnetic T-cell activation beads (Life Technologies, Paisley, UK) were washed three times in complete medium using a Dynal^®^ MPC™-1 magnetic particle concentrator (Life Technologies). The desired volume of washed CD3/CD28 CTS™ paramagnetic bead suspension was added by injection to the MACS cell expansion bag to achieve a bead to cell ratio of 3:1.

Upon completion of these steps, samples of whole blood (remaining after PBMC separation) and PBMC culture were removed for sterility testing using the BactT/ALERT^®^ microbial detection system.

##### Addition of IL-2 to Culture

On day 2, working in a grade D environment, IL-2 (final concentration 100 IU/mL) was added to the culture by sterile welding of the requisite number of pre-aliquoted pouches.

#### 2.1.3. Retroviral Transduction (Day 3; Figures 2C and 3C–F)

On day 3, (48 h post PBMC isolation), activated T-cells were transduced with clinical grade SFG T4 retroviral vector. Vector had been produced under GMP from a master cell bank (BioNTech IMFS, formerly EUFETS, Idar-Oberstein, Germany) and then prepared as 60 mL aliquots in 150 mL Flexboy^®^ bags (Sartorius Stedim, Göttingen, Germany). The master cell bank was extensively tested for adventitious viruses, including replication competent virus. This testing obviated the need for such analysis of the CAR T-cell product since potential sources of adventitious virus were not introduced at other stages during manufacture. In order to further close the manufacturing procedure, bags with weldable tubing could alternatively be used for aliquoting of viral vector.

Working in a grade D environment, a transfer bag was welded to the RetroNectin^®^-coated PermaLife™ bag and unbound RetroNectin^®^ was drained by gravity into the transfer bag, which was then disconnected by sterile tube sealing. Retroviral vector aliquots were stored at −80 °C and one aliquot was rapidly thawed using a Plasmatherm (Barkey, Leopoldshöhe, Germany) according to the manufacturer’s instructions. Thawed vector and the RetroNectin^®^-coated bag were transferred to a grade A isolator, within which the vector was transferred by gravity into the RetroNectin^®^-coated PermaLife™ bag using a custom-made tubing set with MPC and Luer connections (BioPharma Dynamics, Figure 3C). Tubing was disconnected and discarded after use.

All remaining steps were performed in a grade D environment. Activated PBMCs (mainly T-cells) were enumerated using a Scepter™ cell counter. The volume of cell culture required to transfer 20–80 × 10^6^ cells was added to the PermaLife™ bag using the following closed technique. A Luer syringe with tube was welded to the MACS cell expansion bag, allowing removal of the required volume of PBMC (Figure 3D). Next, the Luer syringe with tube was welded to tubing of the PermaLife™ bag, allowing transfer of medium containing activated T-cells (Figure 3E), prior to disconnection by sterile sealing (Figure 3F). Interleukin-2 (100 U/mL, pre-aliquoted in complete medium within a sealed sample pouch) was added to the transduction bag by sterile welding and sealing. The PermaLife bag was now transferred to an incubator (37 °C, 5% CO_2_) and was turned twice at 15-min intervals to allow activated T-cells to engage with both sides of the bag, facilitating gene transfer. Unlike other protocols [12], there was no need for centrifugation of the transduction bag.

#### 2.1.4. Expansion of Transduced Patient-Derived T-Cells (Days 4–14; Figure 2D)

All interventions were performed in a grade D environment. Samples were withdrawn every two days commencing on day 4 by welding sampling pouches to the pre-mixed cell culture bag and are tested for cell count (Scepter cell counter). Based upon this cell count, pre-prepared medium and IL-4 were added by sterile welding, according to rules listed in Table 1. When culture volume exceeded the capacity of the PermaLife™ bag (725 mL), contents were transferred to a MACS^®^ 1000 cell differentiation bag (Miltenyi Biotec) by sterile welding and sealing. Depending on culture volume, transfer to a larger volume MACS^®^ cell differentiation bag was necessary on some occasions. Interim assays assessing the sterility (BactT/ALERT^®^ and mycoplasma PCR) and transduction efficiency were performed on day 8.

#### 2.1.5. Downstream Processing on Final Day of Production (Day 15; Figure 2E)

##### De-Beading of Expansion Culture

On the final or penultimate day of production, the product was de-beaded using a ClinExVivo™ MPC^®^ magnet (Dynal, Life Technologies). Cultures were passed twice over both the primary and secondary magnets through a series of connected transfer bags by gravitational flow. First, the MACS^®^ cell differentiation bag was welded to one of the two tubing lines of a 1000 mL transfer bag. The second line of the transfer bag was welded to a 2000 mL transfer (“collection”) bag. The second line of the 2000 mL transfer bag was sealed. The cell culture bag was hung on the bag stand of the magnet. The 1000 mL transfer bag was placed on the primary magnet bed and the tubing connecting this bag to the 2000 mL transfer bag was coiled around the second magnet. Approximately one third of the culture volume was allowed to pass by gravity to the 1000 mL transfer bag and a clamp applied to stop further flow. After 3 min (to allow for magnetic capture of the beads), the clamp was removed, allowing the culture to pass using gravity to the collection bag via the second magnet. The process was repeated for the remaining two thirds of the cell culture. Next, an aliquot of approximately 200 mL complete medium was welded to the transfer bag placed on the primary magnet bed and this was passed over the retained beads and then into the cell culture to release remaining T-cells that may be loosely associated with the retained beads. To ensure maximal bead removal, the process was repeated using a second set of connected transfer bags.

##### Generation and Release Assay Testing of Drug Substance

The culture (volume up to 2.4 L) was divided equally between two and four 600 mL transfer bags (Terumo) and centrifuged at 300 g for 10 min at room temperature with no brake. The first transfer bag was carefully removed from the centrifuge and hung between the plates of a Plasma Expressor (Genesis BPS, Ramsay, NJ, USA) without dislodging the pellet. A 600 mL transfer bag (waste bag) was welded to the centrifuged bag and the required amount of supernatant was removed after which the bags were separated by sterile sealing. This procedure was repeated with each of the remaining centrifuged bags so that the total volume was approximately 150 mL. A PermaLife™ PL70 (drug substance) bag was sequentially welded to the bags containing the re-suspended cells and cells were transferred into the drug substance bag. Owing to instability of the formulated drug product (see Results section), critical release assays (Table 2) were performed on samples removed from the drug substance by welding of sampling pouches. The drug substance culture was transferred to an incubator (37 °C, 5% CO_2_) while release assays were performed and reviewed by the Qualified Person (QP) for initial certification, a process that was completed in 2.5 h. Initial certification was required before the patient could be treated. It should be noted that transfer of drug substance into bottles used to perform final sterility assays was performed in a grade A isolator. Once these results became available 2–3 weeks later, retrospective final certification of the product was performed by the QP.

##### Formulation of Drug Product

Once release of the cell product was approved, final formulation was initiated in a grade D environment (Figure 4). A cell count was determined (Scepter cell counter) on the drug substance bag. By this means, the final volume was calculated whereby the therapeutic cell dose is contained within the target volume for intra-tumoural injection (1–4 mL). The PermaLife™ bag (volume approximately 150 mL) was centrifuged at 300 g for 10 min at room temperature with no brake (Figure 4A). Excess medium was withdrawn by welding a Luer syringe with tube, which was then disconnected by sterile sealing (Figure 4B). This step was repeated where necessary to allow for the removal of a volume of supernatant greater than 50 mL.

The pelleted cells were re-suspended in the residual medium (Figure 4C). Depending on the final volume, a clamp was placed on the bag to facilitate homogenous re-suspension of the cells. The formulated product was transferred to the administering physician who mixed the bag carefully by repeated inversion (approximately 5–10 inversions). The Luer-Lok connection on the PermaLife™ bag (Figure 4C) was swabbed with an alcohol wipe (70% pharmaceutical grade isopropyl alcohol) and allowed to dry. This port provides a closed, pressure-activated channel through which 1–4 mL volume of the drug product was drawn up by the administering physician. A 21 G needle was then affixed to the syringe and the drug product was administered by the intra-tumoural route, using ultrasound guidance where required.

#### 2.1.6. Alternative T-Cell Activation Process

Owing to supply difficulties with CD3/CD28 CTS™ paramagnetic T-cell activation beads close to the initiation of this trial, the manufacturing process was modified to employ immobilised CD3 and CD28 antibodies (1µg/mL each MACS GMP pure antibodies; Milteny Biotec) as an alternative T-cell activating stimulus. This process is identical to that described above except for the following details. Working under aseptic conditions in a grade A isolator on the day before patient blood is collected, sterile antibodies were pre-coated in a sterile Cell Expansion bag (Miltenyi Biotec) in 50mL of sterile 0.9% saline. The CD3/CD28 antibody-coated bag was stored at 4 °C and was used in Section 2.1.2.

Use of this method obviates the need for steps described in Section 2.1.2 (Preparation and Addition of T-Cell Activation Beads and Section 2.1.5 (De-Beading of Expansion Culture).

### 2.2. Stability Testing

#### 2.2.1. Stability Testing of Cytokines

Cytokines were resuspended in complete medium and aliquoted in sample pouches as indicated in Section 2.1.1. These were stored at 4 °C for 4 days (IL-2) or 14 days (IL-4) as is required in the final manufacturing process. Fresh and stored cytokines (IL-2, 100 U/mL; IL-4 30 ng/mL) or nil was added to three independent cultures of T4^+^ CAR T-cell cultures which had been seeded at 3 × 10^5^ cells/mL in RPMI + 10% human AB serum. To compare function of fresh and stored cytokine aliquots, cultures were counted again after 4 days.

#### 2.2.2. Stability Testing of Drug Product and Drug Substance

To test stability of the drug product, batches of T4 immunotherapy were formulated in 1mL complete medium at two representative densities, namely 1 × 10^7^ cells/mL or 3 × 10^8^ cells/mL. Cells were maintained in a 3 mL Luer syringe at ambient temperature for 2.5 h. This duration was selected as that required for QP certification and release of each CAR T-cell batch. To test stability of the drug substance, batches of T4 immunotherapy were formulated at a density of 8 × 10^6^ cells/mL in a volume of 50 mL complete medium in a PermaLife™ PL70 bag. Cells were maintained in an incubator set to 37 °C and 5% CO_2_ for 2.5 h. In each case, cell count was evaluated in triplicate using a Scepter™ cell counter before and after passing the drug substance or drug product through a 21G needle.

### 2.3. Assessment of Biological Activity of Residual Dynabeads^®^ CD3/CD28

When using CD3/CD28 CTS™ paramagnetic beads in the manufacturing process, there is a concern that residual beads could elicit toxicity owing to unwanted T-cell activation. To assess this risk, Dynabeads^®^ were retrieved at the end of manufacture from two representative batches of T4 immunotherapy and washed three times in complete medium using a Dynal^®^ MPC™-1 magnetic particle concentrator. To compare function of these day 15 beads with freshly prepared beads, PBMCs were isolated from a healthy donor by ficoll density gradient separation and re-suspended in cell culture medium at a concentration of 3 × 10^6^ cells/mL. Fresh or day 15 beads were added at a 3:1 bead:PBMC ratio, making comparison with PBMC alone. Cultures were maintained for 9 days by addition of complete medium + IL-2 (100 UmL) every 2–3 days, after which cell count was evaluated.

### 2.4. Development and Validation of Release Assays

Some critical release assays used for the release of batches of T4 immunotherapy had been developed and validated by the manufacturer (e.g., single platform cell counting using Trucount™ tubes) or by externally contracted organisations (e.g., mycoplasma culture according to the European Pharmacopoeia (Ph. Eur.); Mycoplasma Experience, Reigate, Surrey and 14–21 days direct inoculation testing according to Ph. Eur.; Wickham Laboratories, Gosport, Hampshire, UK). Certificates of analysis were provided by these organisations and were incorporated in the IMPD. Other assays were developed and validated in house as described below.

#### 2.4.1. Purity and Potency of the Drug Product

Purity and potency testing was performed on the drug substance by flow cytometric quantification of CAR expression. Control and T4^+^ T-cells (1 × 10^6^ cells in PBS) were labelled with biotinylated anti-human EGF antibody (BAF236, R&D Systems, Oxon, UK) at a final concentration of 1 μg/mL for 20 min at 4 °C. The sample was washed in 2 mL PBS, centrifuged for 5 min at 200 g, followed by secondary labelling with streptavidin-PE (1 µg; S866, Life Technologies, Paisley, UK) for 20 min at 4 °C. After washing with 2 mL PBS, samples were centrifuged for 5 min at 200 g, re-suspended in 500 μL PBS and analysed using a BD FACSCanto™ II system (Becton Dickinson UK, Wokingham, UK) with FACSDIVA software. Starting cell count and the quantity of both biotinylated anti-human EGF antibody and streptavidin-PE used for the assay were optimised in preliminary experiments (data not shown). To validate this method, T4^+^ T-cells (50 × 10^6^) were labelled as above and were flow sorted to purity using a BD FACSAria™ II Cell Sorter with FACSDIVA software. These cells were mixed with untransduced autologous T-cells at known quantities to create a panel of samples with 0%, 1%, 10%, 25%, 50%, 75% and 100% T4^+^ T-cells. Two investigators independently analysed the panel of samples by flow cytometry in order to quantify the transduced population. Sample analysis was performed using a BD FACSCanto™ II system with FACSDIVA software.

#### 2.4.2. Cell count and Viability Present in the Drug Product

The required number of cells in the drug product is dependent on target dose for each patient. Cell count and viability were quantified using a single platform flow cytometric assay. Using Trucount™ tubes (BD Biosciences, Oxford, UK), 200 µL cells were stained with 5 µg FITC-conjugated anti-human CD45 antibody (HI30, Biolegend, SanDiego, CA, USA) and DAPI (4′,6-diamidino-2-phenylindole; 0.1 µg; Abd Serotec, Kidlington, UK) without washing. Samples were analysed by flow cytometry. The percentage of live CD45^+^ lymphocytes was determined by gating on cells that do not stain with DAPI. TruCount™ beads were enumerated based upon their auto-fluorescence (Figure 5B), allowing the determination of absolute cell count, using the following calculation:$$\frac{\mathrm{No}.\;\mathrm{of}\;\mathrm{events}\;\mathrm{in}\;\mathrm{region}\;\mathrm{containing}\;\mathrm{cells}\;\times \;\mathrm{No}.\;\mathrm{of}\;\mathrm{beads}\;\mathrm{per}\;\mathrm{test}\;=\;\mathrm{absolute}\;\mathrm{cell}\;\mathrm{count}}{\mathrm{No}.\;\mathrm{of}\;\mathrm{events}\;\mathrm{in}\;\mathrm{absolute}\;\mathrm{count}\;\mathrm{bead}\;\mathrm{region}\;\mathrm{test}\;\mathrm{volume}}$$

The gating strategy shown in Figure 6A was used to perform this analysis.

#### 2.4.3. Quantification of Residual CD3/CD28 CTS™ Paramagnetic Beads in Drug Product

Previously, manual counting of residual beads was performed using a haemocytometer [12], a subjective approach that is susceptible to inter-observer variability. We sought to develop a more reproducible and traceable method to quantify residual bead number.

Using flow cytometry, it was observed that CD3/CD28 beads display auto-fluorescence that is maintained throughout cell culture. Samples of T4 immunotherapy (900 μL) were taken before and after de-beading on the final day of production. Cells were lysed in both samples by addition of 10 μL of 10% Triton X-100, followed by vigorous vortexing. An aliquot (500 μL) of each sample was transferred to a Trucount™ tube (BD Biosciences) for analysis. Dynabead® auto-fluorescence was detected in a discrete PerCP^TM^-Cy™5.5 and APC gate using a BD FACSCanto™ II flow cytometer. An APC-Cy™7 threshold was applied to discriminate beads from dead cells and other debris which lacked this autofluorescence. The parameters were set to count 1000 events within the gate encompassing the Trucount™ beads. Residual beads were enumerated using the following formula (adapted from the manufacturer’s instructions):$$\frac{\mathrm{No}.\;\mathrm{of}\;\mathrm{events}\;\mathrm{in}\;\mathrm{region}\;\mathrm{containing}\;\mathrm{CD}3/\mathrm{CD}28\;\mathrm{beads}\;\times \;\mathrm{No}.\;\mathrm{of}\;\mathrm{beads}\;\mathrm{per}\;\mathrm{test}\;=\;\mathrm{absolute}\;\mathrm{bead}\;\mathrm{count}}{\mathrm{No}.\;\mathrm{of}\;\mathrm{events}\;\mathrm{in}\;\mathrm{absolute}\;\mathrm{count}\;\mathrm{bead}\;\mathrm{region}\;\mathrm{test}\;\mathrm{volume}}$$

#### 2.4.4. BacT/ALERT^®^ Sterility Testing

BacT/ALERT^®^ provides an automated microbial detection system in which the test article is inoculated into plastic bottles containing blood culture medium suitable for the propagation of aerobic or anaerobic micro-organisms. Growth of micro-organisms leads to elevation in CO_2_ levels which causes liquid emulsion sensors to change colour, enabling automated colorimetric detection of contamination. A volume of 1mL of test article (T4 immunotherapy harvested on day 8 or 15 of manufacture) was inoculated with saline containing 30 colony-forming units (cfu) of *Pseudomonas aeruginosa* (not mucoid), *Staphylococcus aureus* (not methicillin resistant) or *Aspergillus fumigatus*, making comparison with saline control. Each was inoculated into triplicate aerobic BacT/ALERT^®^ bottles which were incubated for 6 days at 37 °C prior to analysis using standard BacT/ALERT^®^ automated microbial detection systems. Similarly, test article was spiked with 30 cfu of *Clostridium perfringens* and this was inoculated in triplicate into anaerobic bottles. Suspensions of the microbes were calibrated against a McFarland standard 0.5. All culture inoculations were performed by a blinded investigator.

#### 2.4.5. Mycoplasma Detection by Polymerase Chain Reaction

In house mycoplasma detection was performed using the MycoSEQ assay (Life Technologies) according to the manufacturer’s instructions using 100 µL of test article containing a maximum of 1 million cells. Validation was performed by addition of a discriminatory positive control (Life Technologies) in serial dilution from 5–100 copies per test to the test article or water, making comparison with a negative water control. A positive test result required a threshold cycle (C_t_) of <40, melting temperature (T_m_) of 75–8 °C inclusive and derivative value (DV) of ≤0.05.

### 2.5. Statistical Analysis

Data are presented as mean ± SD. Where inter-observer variability was assessed for critical release assays, Bland-Altman pots were generated in order to indicate discrepancy between observers. All statistical analysis was performed using GraphPad Prism version 9.1.0 (San Diego, CA, USA).

## 3. Results

### 3.1. Human AB Serum Is Required for Manufacture of T4 Immunotherapy

Preliminary studies were initially undertaken using PBMC derived from healthy donors in order to define optimum starting materials for manufacture. We confirmed that several GMP-grade media were suitable for this purpose, generating comparable cell yields to research grade media (data not shown). We selected pH indicator-free X-VIVO™ 15 medium for further development. Next, we evaluated the need for human AB serum in the manufacturing process. Exponential expansion and enrichment of T4^+^ T-cells was only observed in IL-4 supplemented cultures (30 ng/mL) that contained 10% AB serum, but not in serum-free medium (Appendix A). A similar requirement for serum was demonstrated with other GMP grade media (data not shown).

### 3.2. Use of Pre-Aliquoted Medium and Cytokines Support Manufacture of T4 Immunotherapy

In order to minimise isolator use throughout manufacture, we next evaluated whether it was possible to pre-aliquot medium and cytokines in a single setting prior to undertaking manufacture. To validate this approach, we undertook stability testing of cytokine-containing complete medium used for this purpose. Appendix A demonstrates that both IL-2 and IL-4 retain full activity when stored for a longer period than is required during a two-week T4 immunotherapy manufacturing run. Maximum determined stability of formulated IL-2 was 4 days and that of IL-4 was 14 days.

### 3.3. Stability Testing of Drug Substance and Drug Product

We planned to administer T4 immunotherapy as a fresh product by intratumoural injection, at densities ranging from 10^7^–10^9^ cells/mL [13]. Consequently, we next tested stability of the drug product for the period required to complete and approve release assay test results (2.5 h). Drug product was formulated at both intermediate and high density and then stored in a Luer syringe at ambient temperature, prior to passage through a 21G needle. This analysis demonstrated that the formulated drug product was unstable over the storage period that would be required for its release by the QP (Figure 5A). When held at lower density, approximately half of the cells were lost in 2.5 h, with even greater cell loss observed at high density. Passage of the cells through a 21 G needle had no additional effect on stability (Figure 5A).

Given the instability of the drug product, we consulted with the UK regulatory authority (Medicines and Healthcare products Regulatory Agency) to discuss the possibility that release testing could instead be performed on the drug substance immediately prior to formulation of the drug product. In order to justify such an approach, we devised a closed process such that formulation of drug product did not entail the addition of any material to the tested drug substance. In brief, this involved centrifugation of the bag containing the drug substance, removal of excess medium by sterile welding/sealing and re-suspension of cells to generate drug product. While release testing was being completed, drug substance was maintained at lower density in complete medium, contained within a gas-breathable bag in an incubator. When maintained in this manner, the drug substance proved to be highly stable (Figure 5B).

### 3.4. Development of Flow Cytometric Release Assays for Cell Count, Viability, Transduction Efficiency and Bead Contamination of Drug Product

Wherever possible, flow cytometric assays were developed to achieve rapid, objective and recordable release assay data. Cell count was enumerated using a single platform assay, incorporating the addition of DAPI to determine viability. Use of Trucount™ tubes to perform single platform cell counting had been validated by the manufacturer. A representative example is shown in Figure 6.

The proportion of transduced T4^+^ T-cells was determined by flow cytometry following antibody staining of the T1E28z CAR. Satisfactory accuracy and inter-observer variability were demonstrated when this analysis was performed by two independent investigators, as demonstrated in Appendix A. All replicate assessments fell within the 95% limits of agreement (Appendix A).

A novel flow cytometric assay was developed to enumerate residual CD3/CD28 CTS™ Dynabeads^®^ in the cell product. Dynabeads^®^ fall within a discrete gate in the forward (FSC-A) v side scatter (SSC-A) plot (Figure 7A). Since Dynabeads^®^ are autofluorescent, a threshold can be set in any one of several fluorescence channels to eliminate non-fluorescent debris (Figure 7B). When Dynabead^®^-containing T-cell cultures are analysed, there is overlap between beads and dead cells (Figure 7C). Following lysis of T-cells with Triton X-100 and setting of a FSC threshold, unacceptable overlap still occurred between Dynabeads^®^ and cell debris (Figure 7D). However, by setting a threshold based on fluorescence in the APC-Cy™7 channel, Dynabeads^®^ could be clearly visualised as a discrete and separate population within a PerCP-Cy™5.5 and APC gate (Figure 7E). Data were normalized to the number of Trucount™ beads counted in a separate gate, allowing the generation of an absolute count. A representative example before and after de-beading is shown in Figure 7F.

In order to establish the accuracy and reproducibility of the assay, samples with known concentrations of Dynabeads^®^ were created by dilution of an original stock from the supplier. Validation of this assay for accuracy and inter-observer variability is shown in Appendix A. Providing further reassurance, residual Dynabeads^®^ present at the end of manufacture proved functionally inactive in assays of T-cell proliferation (Appendix A).

Efficiency of de-beading was only tested in one scale up run, which was performed under full GMP conditions with a bag of clinical grade viral vector. The final bead count was 4000 beads/mL which met the release specification (Table 2). Further testing was not undertaken because of lack of availability of GMP grade Dynabeads^®^ at the time our clinical trial commenced.

### 3.5. Validation of BacT/ALERT^®^ Testing for Sterility of T4 Immunotherapy

Inoculations were confirmed as accurate for a representative aerobe (*P. aeruginosa*), facultative anaerobe (*S. aureus*) and anaerobe (*C. perfringens*). However, the assay failed to detect the representative fungal species (*A. fumigatus*). Consequently, the direct inoculation sterility test was added in order to address this shortcoming, following consultation with the test provider, Wickham Laboratories Ltd. This test involves the introduction of the test article into broth media followed by culture for 14 days. It has been validated for the detection of several micro-organisms and utilises industry standard harmonised methods that are compliant with Ph Eur, USP, JP and ISO standards.

### 3.6. Validation of MycoSEQ Quantitative PCR Test for Mycoplasma Detection

Test article did not alter the sensitivity of this assay for the discriminatory positive control. The assay detected below 10 copies per test as is required by Ph. Eur.

### 3.7. Confirmation of Robustness of Manufacture of T4 Immunotherapy

The manufacturing process described above was iteratively optimised using a series of scale up runs, performed using PBMC from healthy donors (*n* = 13; Figure 8A,B) and patients with HNSCC (*n* = 6; Figure 8C,D). Interleukin-4-dependent enrichment of T4^+^ T-cells occurred in all runs in both normal donor and HNSCC patient cultures (Figure 8B,D). Three engineering runs were conducted using normal donor PBMC under full GMP conditions (shown in red in Figure 8A,B). All three runs were undertaken using the immobilised antibody method of T-cell activation and all met the minimum criteria for release (Table 2). ErbB receptors are expressed on tumour cells and cause T4 CAR T-cells to become activated. Consequently, T-cell products demonstrated cytolytic activity and cytokine release when co-cultured with ErbB-expressing tumor cell targets [8,14,15,16].

## 4. Discussion

Manufacture of patient-specific cellular immunotherapies presents a series of unique challenges that add to costs and impose barriers to large scale production [17]. The complex nature of GMP manufacturing processes coupled with the lack of purity and stability of these living drug products constitute issues that are much less problematic with traditional chemical therapeutics [18], Moreover, marketing authorization requires consistent, robust and reproducible manufacture, supported by rigorous quality control, in addition to the demonstration of safety and efficacy [19,20].

In this manuscript, we describe the approach taken to develop a new sterile manufacturing process for an experimental autologous CAR T-cell therapy for HNSCC. We set out to simplify the manufacturing process while placing emphasis on the development of procedures that maximise the generation of independently recorded and verifiable data.

Automated cell counting was used throughout the process while a number of flow cytometric assays were developed to control aspects of production and enable cell product release. To simplify the procurement of starting material, we elected to use blood rather than leukapheresis product, conscious of the fact that PBMC from individuals with advanced malignancy such as HNSCC are often depleted and poorly functional [21]. In keeping with this, we found that gene transfer efficiency was poorer using cells from patients compared to healthy donors. However, the use of the IL-4/4ab cell expansion system [8,10] reliably promoted enrichment of T4^+^ cells, so that cell products comfortably met release specifications in all cases. Manipulations were performed in a grade D environment using sterile welding/sealing wherever possible. To facilitate this, we devised and validated an approach whereby all medium and cytokines were pre-aliquoted in a grade A isolator. By this means, subsequent culture additions could be performed outside the isolator using sterile welding. We also developed a simplified feeding protocol that was solely based on cell count (determined using welded sampling pouches), without the need for measurement of any metabolic intermediates or pH. Requirement for a grade A isolator use could be eliminated completely if medium constituents, antibodies, RetroNectin^®^, cytokines and viral vector were produced in bags that are amenable to sterile welding. Comparable results were obtained when T-cells were activated using CD3 + CD28-coated Dynabeads^®^ or when these antibodies were pre-immobilised in a cell culture bag. The latter method also obviates the requirement for product de-beading. Alternative GMP-grade T-cell activating materials are also available such as MACS^®^ GMP T Cell TransAct. These considerations have important implications for the manufacture of cell-based therapies in locations where access to advanced GMP facilities and production equipment are limited.

The final drug product does contain a number of potential impurities, including IL-2 and IL-4. Regarding cytokine contamination, the product is concentrated greatly on the final day, reducing risks associated with cytokine transfer. Moreover, both IL-2 and IL-4 have been administered (both systemically and intratumourally) to human subjects in several clinical trials at concentrations that far exceed those that would be present in the drug product.

In advancing this process for future use, a number of modifications could add further benefit. First, our process was semi-manual whereas automation is increasingly recognized to be desirable to maximise standardization and reduce labour [6]. Several closed automated systems are now available or are in development to satisfy this growing need [22]. Second, we elected not to develop a cryopreservation step in the first instance, taking the view that activity of freshly produced cells would be superior to that of a cryopreserved product. Sub-optimal cryopreservation systems are known to compromise recovery, viability, phenotype and function of CAR T-cells [23,24]. However, administration of a fresh CAR T-cell product meant that batch release assays needed to be performed on the day of treatment, a procedure that requires at least 2.5 h. Since the drug product proved unstable when formulated for administration, we developed a strategy to undertake release assay testing on the drug substance. This was justified on the basis that formulation of drug product from drug substance was a fully closed process, undertaken without any additions, meaning that contamination could not theoretically enter the system in this final phase of manufacture. Risk of administration of a contaminated drug product was further mitigated by the requirement that all interim cultures were sterile at the time of treatment. The integrity of this process is indicated by the fact that in all 19 pre-trial production runs, release criteria were met without contamination (confirmed in formal microbiological sterility assays in five cases). Nonetheless, this strategy means that final product certification by the QP must be completed post hoc, once the results of extended sterility assays performed on the drug product became available.

A third intervention that could further enhance product quality entails the use of approaches to enhance cell fitness. We used IL-4 to provide a surrogate IL-2/IL-15 signal to the cells via the 4ab chimeric cytokine receptor, leading to their selective expansion [10]. Further addition of low doses of cytokines such as IL-7 or IL-21 may also warrant investigation in an attempt to delay the differentiation of these cells [25,26]. Alternatively, inhibitors of phosphatidylinositol 3′ kinase, vasoactive intestinal peptide and/or Akt can also inhibit differentiation during manufacture [27,28,29]. While detailed phenotyping of cell products was not undertaken in the development of this manufacturing process, it is now being undertaken in our ongoing Phase 1 clinical trial as a characterisation assay.

At the time of writing, this manufacturing process has been used to successfully treat 16 HNSCC patients in an ongoing Phase 1 clinical trial [13]. Yields were consistently in excess of 1 billion cells from as little as 40mL blood, without any batch failures and achieving target CAR T-cell dose in every case.

## 5. Conclusions

We describe a relatively simple static bag-based CAR T-cell manufacturing process which enables the robust, sterile and reproducible manufacture of panErbB-targeted T-cells from a blood draw. The strategy outlined here is compatible with automation and further iterative improvement through the refinement of culture additions.

## Figures and Tables

**Figure 1 cells-10-01797-f001:**
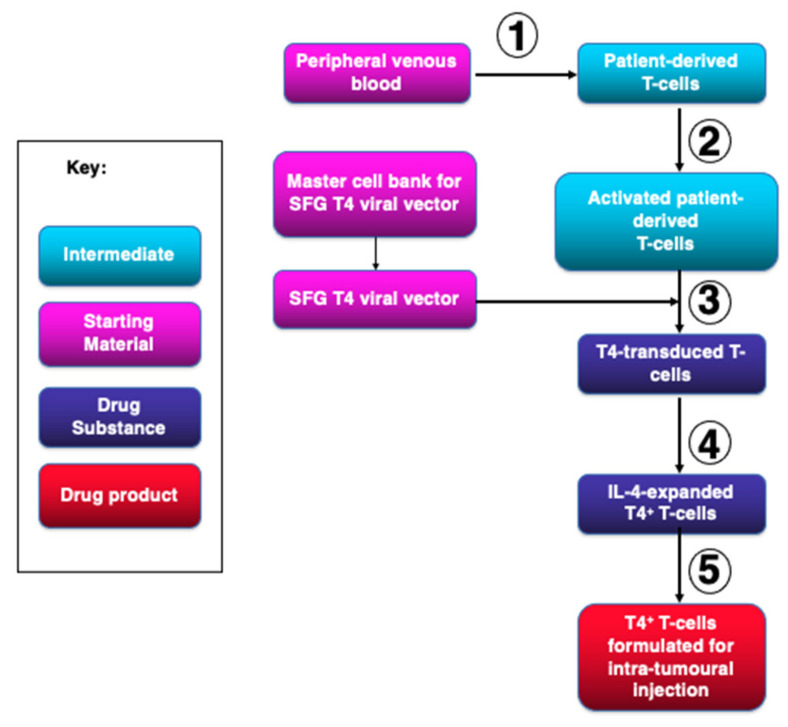
Terminology used in the manufacture of T4 immunotherapy. An overview of the five stages involved in the manufacture of T4 immunotherapy is presented in order to define what constitutes starting materials, drug substance and drug product. Definitions derive from EU Directive 2009/120/EC.

**Figure 2 cells-10-01797-f002:**
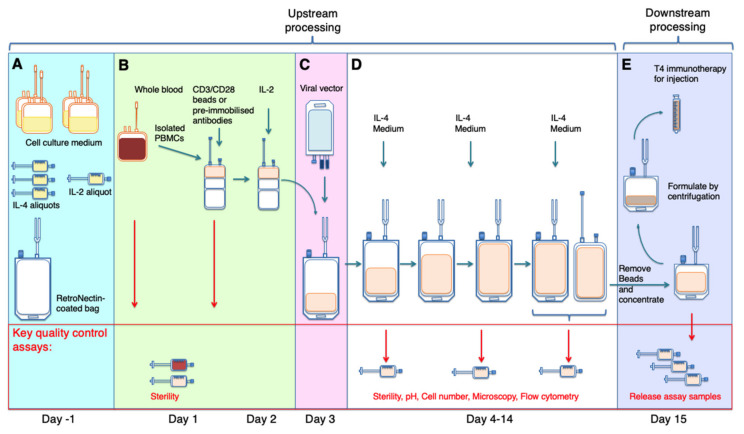
Overview of manufacturing process. (**A**) Sterile medium, RetroNectin^®^ and cytokines were pre-aliquoted in a single setting in a grade A isolator (with rapid vapor hydrogen peroxide sterilisation at a minimum of 100 parts per million for 25 min) on the day before initiating cell product manufacture. (**B**) Patient blood was collected using a closed process into a transfusion bag, which was welded to an automated cell separation platform. Peripheral blood mononuclear cells (PBMC) were isolated, enumerated using a Scepter™ cell counter and transferred by welding to a cell expansion bag. Paramagnetic Dynabeads^®^ CD3/CD28 CTS™ (day 1) and IL-2 (day 2) were added to achieve T-cell activation prior to retroviral transduction on day 3 ((**C**)—shown in greater detail in Figure 3). Transduction may alternatively be undertaken on day 4. (**D**) Thereafter, selective expansion and enrichment of T4^+^ transduced T-cells was achieved by welding of pouches containing recombinant IL-4 (plus medium where required, as specified in Table 1). Samples required for quality control assays are removed by sterile welding of sampling pouches or tube sealing. (**E**) On the final day of production (generally day 15), the drug substance was held in a gas-breathable bag while critical release assays are completed after which it was formulated for injection using a closed system (shown in greater detail in Figure 4). Release of the cell product requires satisfactory completion of interim and final quality control assays. Using this approach, cell product manufacture was achieved with use of a grade A isolator on days −1, 1, 3 and 15 of the process only (approximate total time of 7 h). All steps were conducted within an acceptable temperature range of 15–25 °C.

**Figure 3 cells-10-01797-f003:**
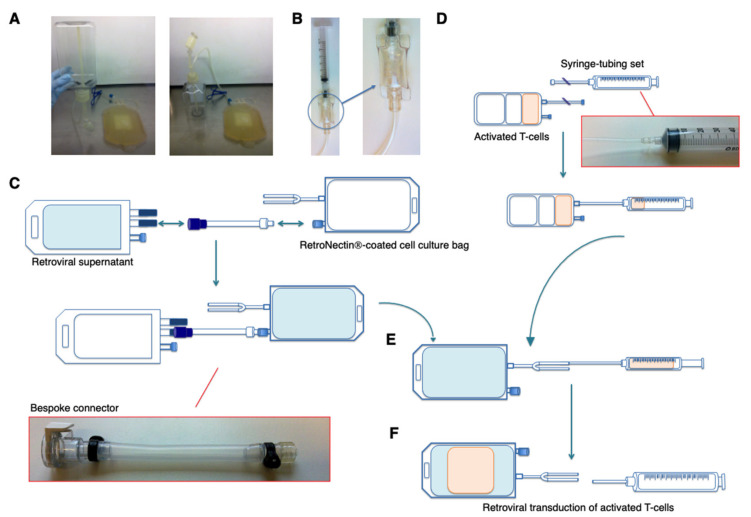
Plasticware used in the manufacture of T4 immunotherapy (**A**) Custom-made lid and tubing set which enables rapid transfer of medium from bottles to transfer bags. (**B**) Luer syringe attached to sample pouch. (**C**) RetroNectin^®^ was removed from a pre-coated PermaLife™ cell culture bag and retroviral supernatant (contained within a Flexboy bag) was transferred using a custom made MPC connection to Luer port tubing set. This transfer was performed in a grade A isolator. The remainder of the transduction process was performed in a grade D environment. (**D**) A Luer syringe with tube was welded to the culture bag containing activated T-cells. The desired volume was withdrawn into the syringe and the tubing sealed to allow for disconnection of the syringe. (**E**) Activated T-cells were introduced to the viral vector-containing PermaLife™ bag by sterile welding and ejection from the syringe. (**F**) The tubing was then sealed and syringe removed.

**Figure 4 cells-10-01797-f004:**
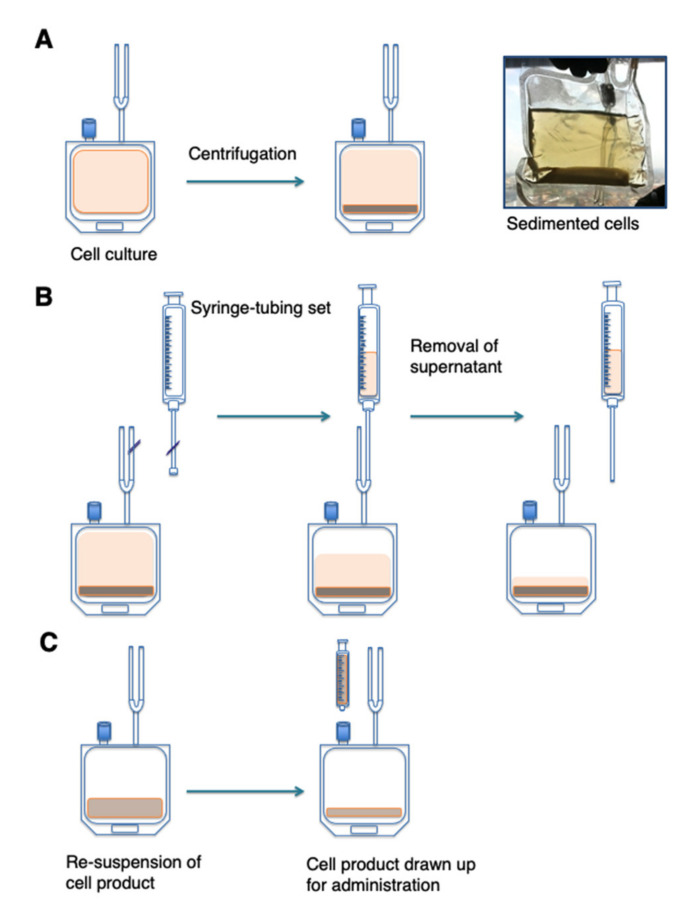
Final formulation of T4 Immunotherapy. (**A**) Prior to final formulation, the culture was sedimented by centrifugation and excess medium was removed using a plasma expressor. Cells were re-suspended in residual medium (volume approximately 150 mL) and held in a cell culture bag in a 5% CO_2_ incubator at 37 °C while the release assays are performed and reviewed. Once certified for clinical use, formulation of the final product was initiated. A cell count is performed (Scepter™ cell counter) in order to calculate the volume for re-suspension after centrifugation, so that the desired dose was contained within 1–4 mL (volume depends on dose to be administered). (**B**) To remove excess supernatant after centrifugation, a Luer syringe with tube was welded to the cell culture bag. Excess supernatant was removed into the syringe, which was disconnected using a tube sealer. Supernatant removed was also tested to ensure that negligible numbers of cells are present. (**C**) Formulation of the product was completed by re-suspending the cells in the residual supernatant, a process that may be assisted by fitting a clamp to the bag (not shown). The bag was transferred to the physician who drew up 1–4 mL of the product for injection by attaching a syringe to the Luer port of the cell culture bag.

**Figure 5 cells-10-01797-f005:**
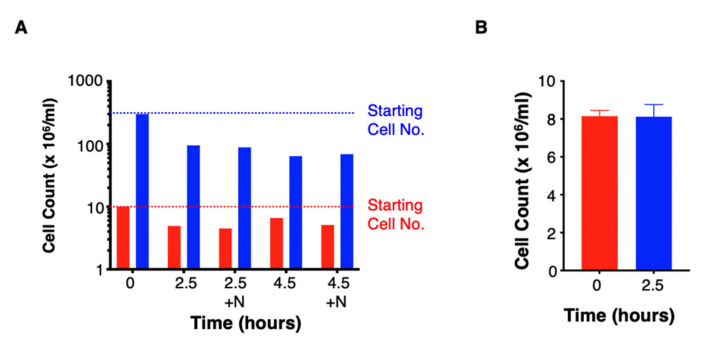
Stability testing of T4 immunotherapy. (**A**) A batch of T4 immunotherapy drug product was formulated in a volume of 1 mL, either at 10^7^ cells/mL or 3 × 10^8^ cells/mL. Cells were suspended in complete medium in a 3 mL Luer syringe with 21 G needle. Cell count was evaluated before and after a holding period of either 2.5 or 4.5 h. In each case, viable cell count was evaluated before or after passage through the needle (+N). (**B**) Drug substance from a batch of T4 immunotherapy was held for 2.5 h at 8 × 10^6^ cells/mL in 50 mL in the final container, which is a gas-breathable cell culture bag. Cells were in complete medium and were held over this period in an incubator set to 37 °C and 5% CO_2_. Data show the mean ± SD of triplicate counts observed before and after this holding period. Data are from a representative experiment. Similar results were obtained in 2 independent experiments.

**Figure 6 cells-10-01797-f006:**
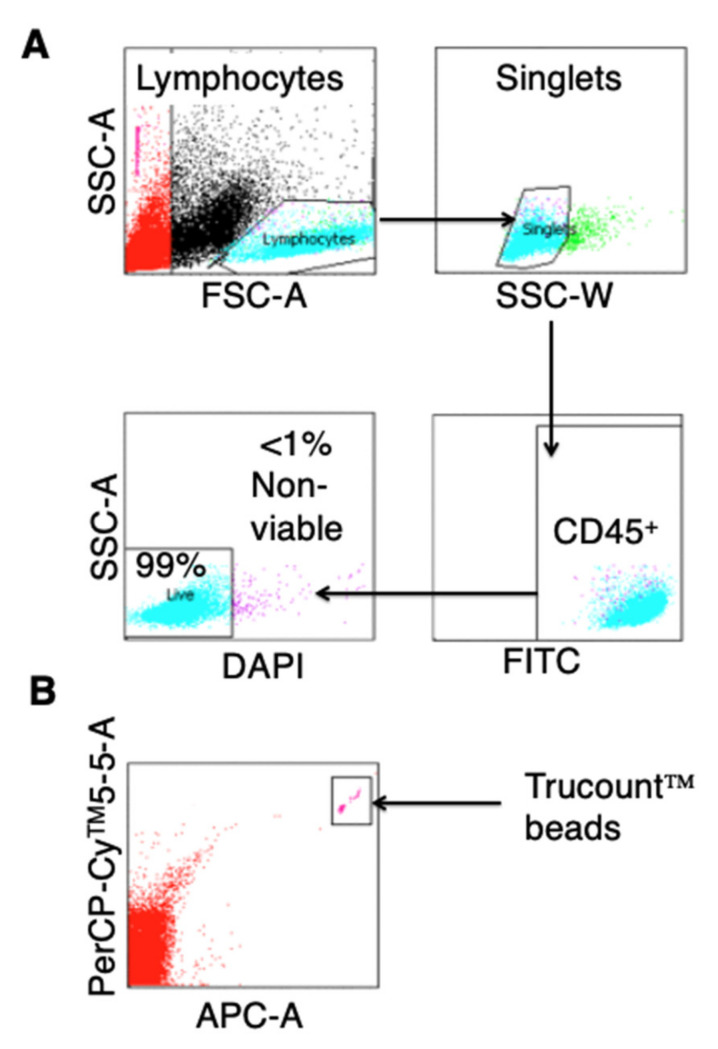
Determination of viability and absolute number of cells present in T4 immunotherapy. Culture samples were incubated in Trucount™ tubes with anti-CD45 and DAPI without washing, facilitating the determination of absolute cell count using a single platform flow cytometric assay. (**A**) Mononuclear cells are first gated using a dot plot of forward scatter pulse area (FSC-A) and side scatter pulse area (SSC-A). By gating on side scatter pulse width (SSC-W), single cells within this population are selected for further analysis. Leukocytes (CD45^+^) are then gated in the FITC channel and used to gate on viable (DAPI^−^) or non-viable (DAPI^+^) cells. (**B**) Data are normalized to the number of Trucount™ beads counted in a separate gate, allowing the generation of an absolute count.

**Figure 7 cells-10-01797-f007:**
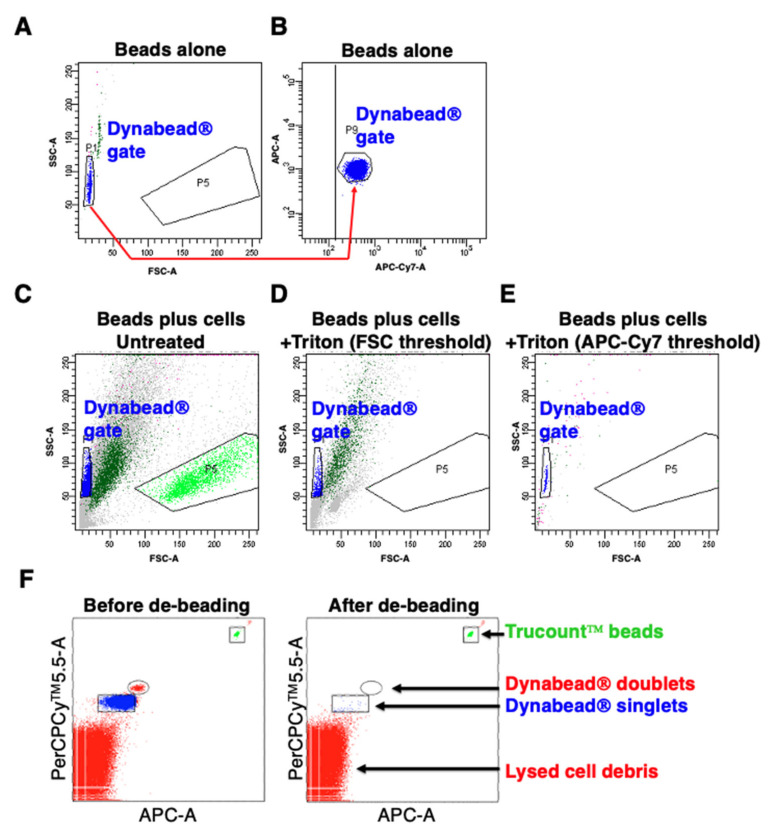
Determination of residual bead number in T4 immunotherapy. CTS^TM^ Dynabeads^®^ fall within a discrete gate in the forward (FSC-A) v side scatter (SSC-A) plot (**A**). A threshold has been set in the APC-Cy^TM^7 fluorescence channel to eliminate non-fluorescent debris (**B**). A representative example in which Dynabead^®^-containing T-cell cultures were analysed without addition of any threshold (**C**). A representative example in which Dynabead^®^-containing T-cell cultures were exposed to Triton X-100 to lyse T-cells. A FSC threshold has been applied (**D**). A representative example in which Dynabead^®^-containing T-cell cultures were exposed to Triton X-100 to lyse T-cells. A threshold has been applied in the APC-Cy^TM^7 fluorescence channel (**E**). (**F**) To enumerate Dynabeads^®^ in cultures prior to and after de-beading, samples were lysed using Triton X-100 and transferred to Trucount™ tubes without washing. Data are normalized to the number of Trucount™ beads counted in a separate gate, allowing the generation of an absolute count.

**Figure 8 cells-10-01797-f008:**
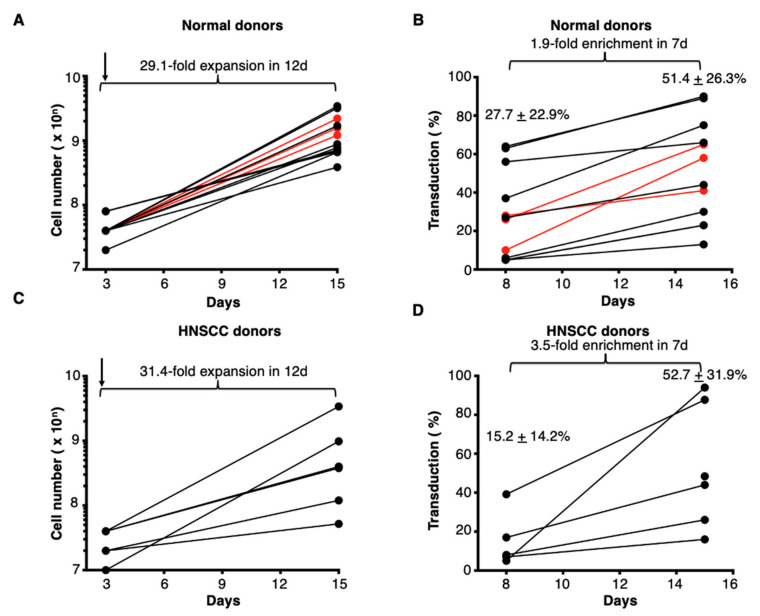
Expansion of T4 immunotherapy under GMP. Activated T-cells derived from healthy donors (**A**,**B**) or patients with head and neck squamous cell carcinoma (HNSCC) (**C**,**D**) were transduced with the T4 retroviral vector 48 h post-isolation (arrowed). Expansion and enrichment of T4^+^ T-cells was achieved by addition of IL-4 as the sole cytokine support. Cell count (**A**,**C**) and % T1E28z^+^ T-cells (**B**,**D**) was determined at the indicated time points. Engineering runs are shown in red in panels A and B and all three were conducted using the immobilised antibody method to achieve T-cell activation. All data show mean ± SD from *n* = 13 (**A**), 12 (**B**), 6 (**C**) or 6 (**D**) donors respectively.

**Table 1 cells-10-01797-t001:** Feeding regimen used during IL-4-mediated expansion of T4^+^ T-cells ^1^.

Cell Concentration (Million Cells per mL)	Medium to Add (% of Existing Culture Volume)
0.5 to 1.0	20%
1.0 to 1.5	100%
>1.5	200%

^1^ Feeding according to these rules was performed every 2 days commencing on day 4 of the manufacturing process. At each feed, the culture was supplemented with additional IL-4 at a final concentration of 30 ng/mL.

**Table 2 cells-10-01797-t002:** Release Specification of the Drug Product.

Analysis	Acceptance Criteria	Method	Notes
Appearance	Clear to opalescent suspension, essentially free from visible particles or particulate matter	Visual Inspection	Performed on bulk Drug Product on the final day of manufacture
Minimum cell number	1 × 10^7^–3.1 × 10^8^ cells	Flow cytometry single platform assay	Minimum number of cells required is in accordance with the cohort for that patient.
Volume	≥2.5 mL (for a dosage volume (DV) of 1 mL) ≥3.5 mL (for a DV of 2 mL) ≥4.5 mL (for a DV of 3 mL) ≥5.5 mL (for a DV of 4 mL)	Volumetric syringe	Supernatant is removed during formulation of the Drug Product. This leaves the correct volume to ensure appropriate dose concentration following re-suspension of the sedimented cells, plus a dead space allowance of 1.5 mL.
Cell viability	≥70%	Flow cytometry single platform assay	Performed on Drug Substance prior to formulation of Drug Product.
Sterility	No growth	BacT/ALERT^®^, direct inoculation testing & Mycoplasma PCR	All In-process control BacT/ALERT^®^ results, and an interim (~1 week) mycoplasma PCR result must be negative for initial certification and release. Final day Drug Substance test articles are taken for BacT/ALERT^®^, mycoplasma culture and Direct Inoculation, and must be negative for full certification.
Identity	Personal identification data on Drug Product label matches exactly with that of the patient.	Verified by two observers	
Purity and potency	≥10% T1E28z^+^ transduced cells	Flow cytometry	Performed on Drug Substance prior to formulation of the Drug Product.
Bead number *	≤333,000 beads per mL	Flow cytometry	Performed on Drug Substance prior to formulation of the Drug Product, allowing calculation of bead concentration in the Drug Product.
Immuno-phenotype	Characterisation assay	Flow cytometry/Time of flight mass cytometry	Detection of markers that include: CD3, CD4, CD8, CD28, CD45RA, CD45RO, CD57, CD62L, CCR7, NKG2D, CD25, CD124, CD19, CD16+56 and PD1

* This proposed limit was not submitted for regulatory approval because the bead activation method was replaced by use of immobilised antibodies.

## Data Availability

Data is available from the corresponding author on request.

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
