# Peer review of "Development and Validation of a Good Manufacturing Process for IL-4-Driven Expansion of Chimeric Cytokine Receptor-Expressing CAR T-Cells"

_cells, 2021, doi:10.3390/cells10071797_

Round 1
Reviewer 1 Report
Major concerns:
- The authors state that they sterilize Isolator grade A using hydrogen peroxide vapor. Materials are assumed to be at least sanitized before entering Isolator grade A, probably in a dedicated pre-compartment for these purposes. However, nowhere is it stated whether the activities carried out within the isolator are environmentally monitored. Do the authors at least use contact plates or air samplers to monitor the presence of viable particles (yeast/mold/bacteria) within the isolator? Do they use settling plates to monitor the procedures that are conducted inside the isolator? The authors must indicate how they ensured that the procedures conducted inside the isolator were free of any viable contaminants.
- Lines 133 and 134 indicate: "Aliquoted media and cytokines are stored at 4°C until use." The authors must indicate at least a time range in which aliquots can be used before they are considered expired. Similarly, this reviewer wants to know if the aliquots were tested for stability at 4°C. If the authors did this, they must indicate it in the manuscript.
- The use of the syringe filled with sterile air is difficult to understand. Is the syringe filled outside, aseptically, from a tank and then inserted into the isolator? Or does the isolator have a sterile air connection and is the syringe filled from the inside? The quality of the air used must be indicated. Do the authors microbiologically monitor the air? Do they monitor the air for the presence of oil? This information is essential to demonstrate the rigor of the bioprocess to satisfy any concerns of a microbiological nature.
- The authors should emphasize the sterility of the process throughout the manuscript. For example, instead of "saline," they should indicate sterile saline. The same holds for PBS, etc. Wherever you have the opportunity to show that they maintained sterile conditions, they should do so.
- The temperature parameter has been ignored in the manuscript. The legend of Figure 2 provides the opportunity to indicate the temperature at which the described processes are carried out. The authors should indicate this important parameter.
- Throughout the manuscript, the terms “cell number” and “cell count” are used interchangeably. Only one of these terms should be used, for consistency. Cell count is recommended.
- In a bioprocess as refined and elegant as the one presented in the manuscript, which has the potential to become the intellectual property of the authors, the use of trademarkTM and registered® symbols absolutely must be used properly. The authors must ensure that the text and the figures have their respective symbols. For example, in Figure 3, RetroNecti lacks the (®) symbol, while in Figure 7, Trucount lacks the (TM)and Dynabeads lacks the (®) symbol. The authors need to make sure to include this information throughout the manuscript, including in the figures and figure legends.
- According to the manufacturer, the BACT/ALERT® system is a simple, automated rapid microbial detection system capable of detecting bacterial, yeast, and mold contamination. In the present manuscript, this identification method was used to identify potential bacteria and fungi present in the product. Did the authors also test for the presence of yeast in the samples? Judging from the controls they used (section 3.5), this does not appear to be the case. A further and very worrying concern is that BACT/ALERT® was not able to identify fumigatus when it was supposed to. The authors’ solution was to contract out the analysis of this type of sample to an outside laboratory, but they do not indicate which method this laboratory used to identify A. fumigatus. They should mention this method and indicate if they had suggested the method employed by the contract lab for this type of bioprocess. Finally, section 2.4.4 should be better explained, as it is hard to follow and the sequence of events is not clear. The authors should provide a better explanation of how BACT/ALERT® works.
- Another critical and worrying element is that the manuscript does not present evidence that the authors performed microbiological quality control tests to determine the presence of adventitious viruses that could contaminate the bioprocess. This type of contamination is well known and very frequent in CAR T cells (Engineering; Volume 5, Issue 1, February 2019, Pages 122-131). The authors must provide evidence that the processes are free of adventitious viruses. In fact, procedures have been developed specifically for this purpose (PDA J Pharm Sci Technol, 68 (6) (2014), pp. 556-562; Vaccine, 34 (17) (2016), pp. 2035-2043 and Vaccine, 32 (52) (2014), pp. 7115-7121). This evidence is critical indeed.
- For many of the procedures presented by the authors, they often mention a welding instrument. What is this instrument? Model? How is sterile welding conducted? Using UV? These details should be mentioned in the manuscript.
- Lines 329 and 330 indicate the following: “Once these results are available 2–3 weeks later, retrospective final certification of the product is performed by the QP.” After this, how long does the product last? Were the authors given an expiration date? Was shelf-life determined?
- Regarding the processes described in section 2.1.5.3: Where are they done? In the isolator? Please clarify.
- Is the 10% human AB serum inactivated? Please include this information in the manuscript.
- Are the bags and tubing used endotoxin free? This must be indicated in the manuscript. In fact, a typical quality test involves an endotoxin test, usually using the LAL method. Did the authors perform this test? This endotoxin information is critical because endotoxins are pyogenic for humans, and they can be present in the product even when the microbiological results are negative. This is because endotoxins, such as LPS, are soluble fragments of the cell walls of the bacteria and not the bacteria themselves.
- Line 628 indicates that no GMP-grade Dynabeads are available. This is a problem because the lack of commercial availability essentially invalidates the process described here. Are there other suppliers? Does the company the authors used continue to manufacture the Dynabeads? Is there any alternative? If so, is it prohibitively expensive?
- Lines 648 and 649 mention the following: “T-cell products demonstrated cytolytic activity and cytokine release when co-cultured with ErbB-expressing tumor cell targets.” Does the use of ErbB confer any inflammatory phenotype to T cells (either pro-inflammatory or anti-inflammatory)? In the manufacturing scenario proposed in this manuscript, are the cytokines removed at any stage? Will they be administered to patients together with T cells? Are any repercussions expected? This, without question, must be discussed in the manuscript.
- Figures 5, 8, and the supplementary figures must indicate n (the number of experiments or replicates).
Minor concerns:
- The Materials and Methods section must be written in the past tense.
- The authors indicate that the pre-production process is done in a grade A Isolator that is sterilized with hydrogen peroxide vapor. However, the concentration of hydrogen peroxide used to sterilize it was not stated. This should be included in the legend of Figure 2.
- This reviewer wants to know why the IL-2 used was not GMP.
- Verify abbreviation of line 188.
- Figure 2 should be improved indicating that panels A, B, C and D belong to the upstream process and panel E is downstream.
- On line 515 the word McFarlane must be corrected, it is McFarland.
- On line 275, the authors should indicate what is the volume necessary to exceed the volume of the bag.
- In lines 315 and 370, the volume must be indicated and the temperature at which the centrifugations were carried out.
- On line 377, you must indicate the approximate number of inversions required.
- On line 378, indicate what type of alcohol was used as well as its concentration.
- In Figure 4, fourth line, the temperature should be indicated and if it was in the presence of CO2 (including the CO2 concentration).
- On line 479, reference is made to the wrong Figure.
- In lines 477 and 478 the final concentration of FITC-conjugated anti-human CD45 antibody and DAPI should be indicated.
Author Response
Reviewer 1
Major concerns:
- The authors state that they sterilize Isolator grade A using hydrogen peroxide vapor. Materials are assumed to be at least sanitized before entering Isolator grade A, probably in a dedicated pre-compartment for these purposes. However, nowhere is it stated whether the activities carried out within the isolator are environmentally monitored. Do the authors at least use contact plates or air samplers to monitor the presence of viable particles (yeast/mold/bacteria) within the isolator? Do they use settling plates to monitor the procedures that are conducted inside the isolator? The authors must indicate how they ensured that the procedures conducted inside the isolator were free of any viable contaminants.
Response: Text in section 2.1. has been modified as follows to address these points as follows:
“A schematic overview of the manufacturing process is provided in Figure 2. Sterile materials and procedures were used throughout the process. Settle (settling) plates were in use throughout manufacturing while operating in a Grade D room. Similarly, settle plates were placed inside a grade A isolator when in use. While operating inside the isolator, the inbuilt alarmed particle counter monitor was turned on. Should the alarm activate, all activities were ceased until the particle count has returned to zero. Finger dabs were performed at the end of each procedure. Where indicated, sterile welding and sealing was performed using a TSCD® Sterile Tubing Welder and T-SEAL® II Tube Sealing Device (Terumo BCT). The wafers used by the device heat up to 300°C, maintaining sterility throughout. All bags, tubing, serum and viral vector used in this procedure were both sterile and endotoxin-free, as specified by the manufacturers.
2.1.1. Pre-production of materials in a grade A isolator (Day -1; Figure 2A)
On the day before blood was collected, medium aliquots, cytokine aliquots and a RetroNectinÒ-coated bag were prepared inside a grade A isolator (Amercare, Thame, UK) that had been sterilized with hydrogen peroxide. All raw materials were sprayed with Klercide™ 70/30 denatured ethanol blended with deionized water, wiped down with Klerwipe™ 70/30 isopropyl alcohol blended with deionized water and stored in a dedicated container prior its use. Upon transfer of raw material to a grade D facility room, raw material was sprayed and wiped a second time as above and then transferred into the grade A isolator via the transfer hatch:”
- Lines 133 and 134 indicate: "Aliquoted media and cytokines are stored at 4°C until use." The authors must indicate at least a time range in which aliquots can be used before they are considered expired. Similarly, this reviewer wants to know if the aliquots were tested for stability at 4°C. If the authors did this, they must indicate it in the manuscript.
Response: Supplementary figure 2 demonstrates the stability testing analysis that was performed on the formulated cytokines. Formulated cytokines were stored at 4°C for the maximum period required in the manufacturing process (e.g. 4 days for IL-2 and 14 days for IL-4). Thereafter the ability of these stored cytokines to stimulate T-cell growth was compared to an equivalent concentration of cytokine that was freshly formulated. This procedure provides a functional evaluation of the cytokines when held at 4°C. We did not test the stability fo these formulated cytokines for longer periods. Consequently, text has been modified as follows: “Maximum determined stability of formulated IL-2 was 4 days and that of IL-4 was 14 days”.
- The use of the syringe filled with sterile air is difficult to understand. Is the syringe filled outside, aseptically, from a tank and then inserted into the isolator? Or does the isolator have a sterile air connection and is the syringe filled from the inside? The quality of the air used must be indicated. Do the authors microbiologically monitor the air? Do they monitor the air for the presence of oil? This information is essential to demonstrate the rigor of the bioprocess to satisfy any concerns of a microbiological nature.
Response: Air inside the isolator is sterile after gassing as proven with air sampling monitoring performed by pharmacy. Consequently, the syringe is filled with air and then connected to the cytokine containing pouch inside the isolator. Air is not routinely monitored for oil. Text in the methods now reads “A 10 ml Luer syringe was filled with sterile air within the grade A isolator and was then attached to the pouch (Figure 3B) to allow transfer of content when pouch is welded to the cell culture.”
- The authors should emphasize the sterility of the process throughout the manuscript. For example, instead of "saline," they should indicate sterile saline. The same holds for PBS, etc. Wherever you have the opportunity to show that they maintained sterile conditions, they should do so.
Response: A sentence has been added in the methods section which reads “Sterile materials and procedures were used throughout the process.” In addition, the word sterile has been inserted throughout the manuscript.
- The temperature parameter has been ignored in the manuscript. The legend of Figure 2 provides the opportunity to indicate the temperature at which the described processes are carried out. The authors should indicate this important parameter.
Response: We have added a sentence to the legend in Figure 2 which reads “All steps are conducted within an acceptable temperature range of 15-25°C.
- Throughout the manuscript, the terms “cell number” and “cell count” are used interchangeably. Only one of these terms should be used, for consistency. Cell count is recommended.
Response: Cell count has been substituted for cell number.
- In a bioprocess as refined and elegant as the one presented in the manuscript, which has the potential to become the intellectual property of the authors, the use of trademarkTM and registered® symbols absolutely must be used properly. The authors must ensure that the text and the figures have their respective symbols. For example, in Figure 3, RetroNecti lacks the (®) symbol, while in Figure 7, Trucount lacks the (TM)and Dynabeads lacks the (®) symbol. The authors need to make sure to include this information throughout the manuscript, including in the figures and figure legends.
Response: We thank the reviewer for this kind comment and have corrected these deficiencies.
- According to the manufacturer, the BACT/ALERT® system is a simple, automated rapid microbial detection system capable of detecting bacterial, yeast, and mold contamination. In the present manuscript, this identification method was used to identify potential bacteria and fungi present in the product. Did the authors also test for the presence of yeast in the samples? Judging from the controls they used (section 3.5), this does not appear to be the case. A further and very worrying concern is that BACT/ALERT® was not able to identify fumigatus when it was supposed to. The authors’ solution was to contract out the analysis of this type of sample to an outside laboratory, but they do not indicate which method this laboratory used to identify A. fumigatus. They should mention this method and indicate if they had suggested the method employed by the contract lab for this type of bioprocess. Finally, section 2.4.4 should be better explained, as it is hard to follow and the sequence of events is not clear. The authors should provide a better explanation of how BACT/ALERT® works.
Response: A brief description of the BacT/ALERTÒ system has been provided in the methods as follows: “BacT/ALERTÒ provides an automated microbial detection system in which the test article is inoculated into plastic bottles containing blood culture medium suitable for the propagation of aerobic or anaerobic micro-organisms. Growth of micro-organisms leads to elevation in CO2 levels which causes liquid emulsion sensors to change colour, enabling automated colorimetric detection of contamination.”
We did not validate the BACT/ALERT® system for the detection of yeast and as correctly identified by the reviewer, it did not perform well in the detection of A. fumigatus. To address the point regarding the externally contracted sterility test, the following text has been added “Consequently, the direct inoculation sterility test was added in order to address this shortcoming, following consultation with the test provider, Wickham Laboratories Ltd. This test involves the introduction of the test article into broth media followed by culture for 14 days. It has been validated for the detection of several micro-organisms and utilises industry standard harmonised methods that are compliant with Ph Eur, USP, JP and ISO standards.”
- Another critical and worrying element is that the manuscript does not present evidence that the authors performed microbiological quality control tests to determine the presence of adventitious viruses that could contaminate the bioprocess. This type of contamination is well known and very frequent in CAR T cells (Engineering; Volume 5, Issue 1, February 2019, Pages 122-131). The authors must provide evidence that the processes are free of adventitious viruses. In fact, procedures have been developed specifically for this purpose (PDA J Pharm Sci Technol, 68 (6) (2014), pp. 556-562; Vaccine, 34 (17) (2016), pp. 2035-2043 and Vaccine, 32 (52) (2014), pp. 7115-7121). This evidence is critical indeed.
Response: Adventitious virus contamination is a risk that is linked to materials used to generate viral vector. This is acknowledged in the cited study in Engineering which states that “The key raw materials commonly used for vector production are cells, media and serum, and plasmids. Each of these must come from an approved supplier and should undergo rigorous testing procedures to reduce the risk of introducing adventitious factors into the production process. A cell bank system is required for the retrovirus packaging for stably transfected cells.”
In our study, testing for adventitious viruses (including replication competent retrovirus) was performed on the viral vector master cell bank as part of this externally contracted work. Given that testing was negative, it was not deemed necessary to perform such testing on CAR T-cell products. Test in the methods section has been added to indicate this point as follows: “The master cell bank was extensively tested for adventitious viruses, including replication competent virus. This testing obviated the need for such analysis of the CAR T-cell product since potential sources of adventitious virus were not introduced at other stages during manufacture.”
- For many of the procedures presented by the authors, they often mention a welding instrument. What is this instrument? Model? How is sterile welding conducted? Using UV? These details should be mentioned in the manuscript.
Response: A sentence has been added to the methods section which reads “Where indicated, sterile welding and sealing was performed using a TSCD® Sterile Tubing Welder and T-SEAL® II Tube Sealing Device (Terumo BCT). The wafers used by the device heat up to 300°C, maintaining sterility throughout.”
- Lines 329 and 330 indicate the following: “Once these results are available 2–3 weeks later, retrospective final certification of the product is performed by the QP.” After this, how long does the product last? Were the authors given an expiration date? Was shelf-life determined?
Response: The product is administered within 2 hours of formulation. This means that final certification by the QP takes place after the product has been administered to the patient. A sentence has been added to section 2.1.5.2 to indicate that “Initial certification was required before the patient could be treated.”
- Regarding the processes described in section 2.1.5.3: Where are they done? In the isolator? Please clarify.
Response: This has been clarified with the following edited introductory sentence to section 2.1.5.3. “Once release of the cell product is approved, final formulation was initiated in a grade D environment (Figure 4).”
- Is the 10% human AB serum inactivated? Please include this information in the manuscript.
Response: Serum has not been heat inactivated. A sentence in section 2.1.1 has been modified to read “To prepare complete medium, 10% human AB serum (non-heat inactivated; Seralab, Haywards Heath, UK) and 10% stable glutamine ATMP ready (PAA, Yeovil, UK) are added to four 1L bottles of X-VIVOÔ 15 medium (BE02-054Q; Lonza, Slough, UK).”
- Are the bags and tubing used endotoxin free? This must be indicated in the manuscript. In fact, a typical quality test involves an endotoxin test, usually using the LAL method. Did the authors perform this test? This endotoxin information is critical because endotoxins are pyogenic for humans, and they can be present in the product even when the microbiological results are negative. This is because endotoxins, such as LPS, are soluble fragments of the cell walls of the bacteria and not the bacteria themselves.
Response: Endotoxin testing was not performed during CAR T-cell manufacture. Vector, serum, bags and tubing were all certified as endotoxin free. A sentence has been added (line 114-5) which reads “All bags, tubing, serum and viral vector used in this procedure are endotoxin-free, as specified by the manufacturers.”
- Line 628 indicates that no GMP-grade Dynabeads are available. This is a problem because the lack of commercial availability essentially invalidates the process described here. Are there other suppliers? Does the company the authors used continue to manufacture the Dynabeads? Is there any alternative? If so, is it prohibitively expensive?
Response: At the time our trial commenced, we were unable to source these beads. However, they are now available for non-commercial manufacture. In our manuscript, we present an alternative activation protocol using immobilised antibodies. Other activating reagents are also available such as TransActTM (Miltenyi Biotec). We have added a sentence to indicate that the supply issue with these beads was temporary “Further testing was not undertaken because of lack of availability of GMP grade DynabeadsÒ at the time our clinical trial commenced.” In the Discussion, we have added the following sentence “Alternative GMP-grade T-cell activating materials are also available such as MACS® GMP T Cell TransAct.”
- Lines 648 and 649 mention the following: “T-cell products demonstrated cytolytic activity and cytokine release when co-cultured with ErbB-expressing tumor cell targets.” Does the use of ErbB confer any inflammatory phenotype to T cells (either pro-inflammatory or anti-inflammatory)? In the manufacturing scenario proposed in this manuscript, are the cytokines removed at any stage? Will they be administered to patients together with T cells? Are any repercussions expected? This, without question, must be discussed in the manuscript.
Response: text has been added to the revised manuscript to address these two points as follows.
Line 690 - ErbB receptors are expressed on tumour cells and cause T4 CAR T-cells to become activated. Consequently, T-cell products demonstrated cytolytic activity and cytokine release when co-cultured with ErbB-expressing tumor cell targets [8, 14-16]
Line 759 - The final drug product does contain a number of potential impurities, including IL-2 and IL-4.
Regarding cytokine contamination, the product is concentrated greatly on the final day, reducing risks associated with cytokine transfer. Moreover, both IL-2 and IL-4 have been administered (both systemically and intratumourally) to human subjects in several clinical trials at concentrations that far exceed those that would be present in the drug product.
- Figures 5, 8, and the supplementary figures must indicate n (the number of experiments or replicates).
Response: This information has been added to the specified figures.
Minor concerns:
- The Materials and Methods section must be written in the past tense.
Response: This has been corrected.
- The authors indicate that the pre-production process is done in a grade A Isolator that is sterilized with hydrogen peroxide vapor. However, the concentration of hydrogen peroxide used to sterilize it was not stated. This should be included in the legend of Figure 2.
Response: The figure legend has been modified to provide this information as follows: “Sterile medium, RetroNectinÒ and cytokines were pre-aliquotted in a single setting in a grade A isolator (with rapid vapor hydrogen peroxide sterilisation at a minimum of 100 parts per million for 25 minutes) on the day before initiating cell product manufacture.”
- This reviewer wants to know why the IL-2 used was not GMP.
Response: Aldesleukin is pharmaceutical grade IL-2 which is suitable for administration to humans, meaning that it is suitable for incorporation into a GMP manufacturing process.
- Figure 2 should be improved indicating that panels A, B, C and D belong to the upstream process and panel E is downstream.
Response: This has been corrected.
- On line 515 the word McFarlane must be corrected, it is McFarland.
Response: This has been corrected.
- On line 275, the authors should indicate what is the volume necessary to exceed the volume of the bag.
Response: This has been added (725mL).
- In lines 315 and 370, the volume must be indicated and the temperature at which the centrifugations were carried out.
Response: These sentences have been amended to include this information as follows:
The culture (volume up to 2.4L) was divided equally between two and four 600 mL transfer bags (Terumo) and centrifuged at 300g for 10 minutes at room temperature with no brake.
The PermaLifeÔ bag (volume approximately 150mL) was centrifuged at 300g for 10 minutes at room temperature with no brake (Figure 4A).
- On line 377, you must indicate the approximate number of inversions required.
Response: This sentence has been amended to include this information as follows:
The formulated product was transferred to the administering physician who mixed the bag carefully by repeated inversion (approximately 5-10 inversions).
- On line 378, indicate what type of alcohol was used as well as its concentration.
Response: This sentence has been amended to include this information as follows:
The Luer-Lok connection on the PermaLifeÔ bag (Figure 4C) was swabbed with an alcohol wipe (70% pharmaceutical grade isopropyl alcohol) and allowed to dry.
- In Figure 4, fourth line, the temperature should be indicated and if it was in the presence of CO2 (including the CO2 concentration).
Response: This sentence has been amended to include this information as follows:
Cells are re-suspended in residual medium (volume approximately 150mL) and held in a cell culture bag in a 5% CO2 incubator at 37°C while the release assays are performed and reviewed.
- On line 479, reference is made to the wrong Figure.
Response: This has been corrected.
- In lines 477 and 478 the final concentration of FITC-conjugated anti-human CD45 antibody and DAPI should be indicated.
Response: This sentence has been amended to include this information as follows:
Using TrucountÔ tubes (BD Biosciences, Oxford, UK), 200µL cells were stained with 5µg FITC-conjugated anti-human CD45 antibody (HI30, Biolegend, SanDiego, CA) and DAPI (4',6-diamidino-2-phenylindole; 0.1µg; Abd Serotec, Kidlington, UK) without washing.
Reviewer 2 Report
This article is overally well organized to show CAR-T manufacturing process pursuing further improvements including automation and the refinement of culture additions. I think this is suitable for publication in the journal Cells.
Author Response
Reviewer 2
This article is overally well organized to show CAR-T manufacturing process pursuing further improvements including automation and the refinement of culture additions. I think this is suitable for publication in the journal Cells.
Response: We thanks the reviewer for these kind comments.
Reviewer 3 Report
The manuscript provided by van Schalkwyk et al. deals with the manufacturing process development for CAR-T therapy. It contains technically useful information for researchers working in this field. However, the scientific findings and insights are weak. Therefore, it would be appropriate to submit to another journal in the specialized field.
Author Response
Reviewer 3
The manuscript provided by van Schalkwyk et al. deals with the manufacturing process development for CAR-T therapy. It contains technically useful information for researchers working in this field. However, the scientific findings and insights are weak. Therefore, it would be appropriate to submit to another journal in the specialized field.
Response: We thank the reviewer for this observation. It was the lack of technical descriptions of CAR T-cell manufacturing processes that prompted us to submit this manuscript to a Cells special issue entitled “Cell and Gene Therapy of Cancer”. We hope that this will enable access to a readership with an appropriate specialist interest.
Round 2
Reviewer 3 Report
No comment on the manuscript